

# Changes in regional climate extremes as a function of global mean temperature: an interactive plotting framework

Richard Wartenburger[1], Martin Hirschi[1], Markus G. Donat[2,3], Peter Greve[4], Andy J. Pitman[2,3], and Sonia I. Seneviratne[1]

[1]Institute for Atmospheric and Climate Science, ETH Zurich, Zurich, Switzerland
[2]ARC Centre of Excellence for Climate System Science, University of New South Wales, Sydney, Australia
[3]Climate Change Research Centre, University of New South Wales, Sydney, Australia
[4]International Institute for Applied Systems Analysis (IIASA), Laxenburg, Austria

*Correspondence to:* Richard Wartenburger (richard.wartenburger@env.ethz.ch), Sonia I. Seneviratne (sonia.seneviratne@ethz.ch)

**Abstract.** This article extends a previous study (Seneviratne et al., 2016) to provide regional analyses of changes in climate extremes as a function of projected changes in global mean temperature. We introduce the DROUGHT-HEAT Regional Climate Atlas, an interactive tool to analyse and display a range of well-established climate extremes and water-cycle indices and their changes as a function of global warming. These projections are based on simulations from the $5^{th}$ phase of the Coupled Model Intercomparison Project (CMIP5). A selection of example results are presented here, but users can visualize specific indices of interest using the online tool. This implementation enables a direct assessment of regional climate changes associated with global temperature targets, such as the 2 degree and 1.5 degree limits agreed within the 2015 Paris Agreement.

## 1 Introduction

The 2015 United Nations Climate Change Conference in Paris (COP21) recently set the goal of limiting global temperature increases to "well below 2 degrees" and to pursue efforts to limit warming to 1.5 °C above pre-industrial levels. Despite this global agreement, the implications of these global temperature thresholds have not been fully assessed. Specifically, stakeholders, decision makers, and the public need more detailed information with respect to associated changes on regional scales, in particular for extreme events and impacts on humans and ecosystems (e.g. Seneviratne et al., 2016, hereafter S16; see also e.g. Schleussner et al., 2016, Guiot and Cramer, 2016, James et al., 2017).

Numerous approaches have recently been developed for identifying regional climate signals associated with specific global warming targets (James et al., 2017). The technique used in S16 and this study is an empirical sampling approach, which contrarily to pattern scaling (e.g., Huntingford and Cox, 2000; Mitchell, 2003; Tebaldi and Arblaster, 2014; Lopez et al., 2014), does not require a priori assumptions on the dependency on global temperature (or other climate variables, e.g. Frieler et al., 2012; Lynch et al., 2017 (in review); Kravitz et al., 2017 (in review)). Namely, the approach used in S16 derives for predefined regions the empirical dependency of changes in regional quantities (e.g. extremes or mean of climate variables, possibly also impacts (S16)) as a function of global temperature changes based on a range of climate model projections. This





approach can be viewed as an "empirical global temperature sampling" technique, which is a type of hybrid approach compared to the four main approaches described in James et al., 2017. S16 has shown that for some extremes (annual maximum and minimum temperature, heavy precipitation events), the ensemble mean response of absolute changes was often found to be linear, consistent with assumptions of some of the pattern scaling literature and results from other publications (e.g., Fischer

et al., 2014). However, this approach also allows to visually assess non-linearities in the relationships.

We provide an illustration of the display used in S16 in Fig. 1. The main advantage of this approach is that it provides in a single figure information on a) the response of a given regional quantity for different global temperature (and greenhouse gas emissions) targets, b) an empirical assessment of this relationship (allowing e.g. to identify its possible (non-) linearity), and c) the range of model and scenario response around this value. Hence, complex information can be more easily conveyed to

regional stakeholders, instead of being summarized in several global analyses or provided as a time- and scenario-dependent information. While globally aggregated information also has obvious value (e.g., O'Neill et al., 2017), regional information is of critical importance for adaptation and communication.

The S16 study, which focused on temperature and precipitation extremes for two emissions scenarios (RCP8.5 and RCP4.5), identified that much of the absolute changes in temperature extremes and heavy precipitation events could be related almost

linearly to the changes in global mean temperature for the time period 1860–2099 (see also Fig. 1), and that this relationship was very similar for the two different emissions scenarios. In addition, it highlighted that – in absolute terms – changes in regional temperature extremes tended to be much larger than the global mean temperature change. The regional model spread was found to be highly variable depending on the considered quantity and region (S16). We note that all analyses focused on the transient climate response, and not on the response at climate equilibrium, which is expected to be substantially

different. In addition, it does not consider aspects related to e.g. overshooting of climate targets or irreversibility in the climate response (Knutti et al., 2016). Moreover, S16 considered changes in absolute temperature extremes and not in the exceedance of given temperature thresholds, which by design would tend to change exponentially when mean regional temperature approaches the set threshold (e.g., Fischer and Knutti, 2015), even in the case of a linear dependency of the changes in absolute temperature extremes (Whan et al., 2015).

As a follow-up to the S16 study, we provide several new contributions and analyses. First, we introduce a new web-based interactive plotting framework (hereafter referred to as the "DROUGHT-HEAT Regional Climate Atlas", available via the URL http://www.drought-heat.ethz.ch/atlas) for the visualization of key relationships, so that the results can be easily shared with other researchers and stakeholders. The DROUGHT-HEAT Regional Climate Atlas has been augmented by several variables compared to the analyses of S16, including responses in regional mean temperature and precipitation and additional

climate extremes. In addition, the analyses are performed for all four CMIP5 emissions scenarios (RCP2.6, RCP4.5, RCP6.0, and RCP8.5). These results can be assessed interactively by users on-line. An overview of the main relationships and a comparison with the previous analyses of S16 are discussed in Sect. 3.1. We provide some detailed analyses of specific features of interest for the interpretation of the results. In particular, we assess differences in regional responses at 1.5 °C, 2 °C and 3 °C global temperature increases in Sect. 3.2. We also assess differences between intra-model spread (i.e. from several realisations

of the same model) and inter-model spread for the derived relationships in Sect. 3.3. Finally, we provide analyses for regional





Changes in extremes on y-axis: can be used to determine critical global temperature thresholds from a regional perspective

Total model range (minimum to maximum response)

Multi-model mean response

Identity line (for temperature indices): Indicates whether regional extremes warm more or less than the global temperature

**Figure 1.** Example of a plot displaying the empirical global temperature dependency of a regional climate index following the S16 approach, including explanatory annotations (adapted from S16).

mean temperature and precipitation based on simulations beyond 2100 (section 3.4), to assess the links between long-term vs. short-term responses.

## 2   Methods and data

This section presents the data sources and methods used to produce the DROUGHT-HEAT Regional Climate Atlas. It is structured as follows: Sections 2.1 and 2.2 introduce the set of model simulations and climate and extremes indices which the analyses are based on. The S16 empirical global temperature sampling approach is presented in Sect. 2.3. Finally, Sect. 2.4 describes the content and technical implementation of the DROUGHT-HEAT Regional Climate Atlas.



## 2.1 Model simulations

The presented regional-scale dependencies between global mean temperature and a range of indices are derived from global climate model (GCM) simulations from the Coupled Model Intercomparison Project Phase 5 (CMIP5, Taylor et al., 2012). The subset of GCMs used in this study includes all models for which a) daily data is available within CMIP5 and b) for

which climate change indices from the joint CCl/CLIVAR/JCOMM Expert Team on Climate Change Detection and Indices (ETCCDI) are available (Sillmann et al., 2013a, b).

To assess the impact of intra-model spread, we perform our analysis in two steps: using a) only one ensemble member per model (r1i1p1), and b) all members available. Similar to S16, we focus on model simulations over the time period 1861–2099, as this is the period covered by virtually all models. For the evaluation of the global temperature dependency beyond the end

of the century, we also analyse a subset of simulations spanning all years from 1861 till 2299. For clarity of visual display, we excluded model simulations of the RCP8.5 scenario for which no simulations exist in the historical period. To facilitate the calculation of regional ensemble averages, all GCM output has been bi-linearly interpolated to a horizontal resolution of $2.5°\text{x}2.5°$. The final set of model simulations employed in this study is listed in Table 1.

## 2.2 Climate and extremes indices

For the ensemble member $e$ of each model $m$ and emission scenario $rcpx$, we have analysed the 27 ETCCDI core climate change indices $I_{rcpx,m,e}$, which were downloaded from the Canadian Centre for Climate Modelling and Analysis (CCMA) indices archive (http://www.cccma.ec.gc.ca/data/climdex/; Sillmann et al., 2013a, b) on 19 May 2016. Similar to the CMIP5 model data, the indices have been interpolated to $2.5°\text{x}2.5°$ horizontal resolution.

In addition to the ETCCDI indices, we have computed three drought indices (which can be used to monitor either anoma-

20 lously dry or anomalously wet conditions) based on soil moisture, precipitation and evapotranspiration from CMIP5 model simulations (see Sect. 2.1) using the R statistical language and the Climate Data Operators (CDO). The Standardized Precipitation Index ($SPI$) has been calculated using the SPEI package (https://cran.r-project.org/web/packages/SPEI, based on Vicente-Serrano et al., 2010) for an accumulation period of 12 months. Soil moisture anomalies ($SMA$, given in units of standard deviations in order to be independent on model-specific parametrisations of soil moisture depths) have been derived

according to the procedure used in Orlowsky and Seneviratne (2012, 2013), which includes a posterior filtering of $SMA$ using a median absolute deviation filter. In addition, we provide analyses for changes in precipitation minus evapotranspiration ($P - E$), as a further measure of changes in land water availability (e.g., Greve and Seneviratne, 2015).

We also include mean temperature ($T$) and precipitation ($P$) in our analyses. We do this to assess whether the regional response of extremes is related to the regional mean climate response or rather reflects a specific behaviour of extremes in the

30 regions examined. For simplicity, we also refer to these variables as indices. A complete list of all indices, their data source and associated units is provided in Table 2.



## 2.3 Derivation of regional global temperature dependency relationships

Yearly global mean temperatures $T_{glob,rcpx,m,e}$ for emission scenario $rcpx$ have been derived from each ensemble member $e$ of model $m$. Both $I_{rcpx,m,e}$ and $T_{glob,rcpx,m,e}$ are treated as anomalies relative to the pre-industrial reference period 1861–1880 (subscript $ref$). For all time steps $t$, we thus compute: $\Delta T_{glob,rcpx,m,e,t} = T_{glob,rcpx,m,e,t} - T_{glob,rcpx,m,e,ref}$ and

$\Delta I_{rcpx,m,e,t} = I_{rcpx,m,e,t} - I_{rcpx,m,e,ref}$. Note that $T_{glob}$ refers to a model estimate of past and predicted future global mean near-surface temperatures which is known to be biased with respect to observation-based global mean temperature records that merges air temperatures over land and sea surface temperatures over the ocean (Cowtan et al., 2015).

We apply a common land-sea mask at $2.5° \text{x} 2.5°$ to all indices as we focus on (extremes) indices that are meaningful over land. We then compute regionally averaged indices $\Delta I_{reg,rcpx,m,e}$ using the set of globally distributed regions defined

in Chapter 3 of the Special Report on Managing the Risks of Extreme Events and Disasters to Advance Climate Change Adaptation (SREX, Seneviratne et al., 2012, Fig. 3-1 therein), hereafter referred to as SREX regions. We also average the indices over the additional regions defined in S16 as well as over global land (including ice sheets).

To test the significance of the dependency relationship between the global temperature signal and the regionally averaged indices, we apply an ordinary least squares fit between $\Delta T_{glob,rcpx,m,e}$ and $\Delta I_{reg,rcpx,m,e}$ for each individual model realization

(focusing on $\Delta T_{glob,rcpx,m,e} \geq 1°C$, which roughly represents future projections in the individual model simulations). The number of models for which the dependency relationship approximated by a linear regression is significantly different from zero ($p = 0.01$, after controlling the false discovery rate according to Benjamini and Hochberg, 1995, as recently suggested by Wilks, (in press)) is used to indicate the robustness of the relationship in the ensemble mean of the changes (see Sect. 3). Note that a significant response of an individual model realization implies that the corresponding dependency relationship

can be explained by a linear model, though it does not guarantee superiority of the linear model over other, higher-order polynomials.

To filter out short-term climatic fluctuations, a decadal running mean is applied to the anomalies starting with 1871–1880 (note that the year associated with each running mean period refers to the last year of that period). We then compute the unweighed ensemble mean change of the smoothed indices $\Delta I_{reg,rcpx} = \overline{\Delta I_{reg,rcpx,m,e}}$ and the corresponding ensemble

mean change of the global mean temperatures $\Delta T_{glob,rcpx} = \overline{\Delta T_{glob,rcpx,m,e}}$.

In order to yield common, model-independent values of $\Delta T_{glob}$ and to provide a bidirectional uncertainty estimate (i.e., including both the inter-model ensemble spread in $\Delta I_{reg,rcpx}$ and $\Delta T_{glob,rcpx}$), we perform a spline interpolation of $\Delta I_{reg,rcpx,m,e}$ to a common temperature axis. The minimum and maximum of the interpolated values (across all model realizations and scenarios) are then used to determine the overall spread of $\Delta I_{reg}$ relative to $\Delta T_{glob}$.



### 2.4 Plotting framework

#### 2.4.1 Content of the plotting framework

All regional dependency plots and related figures similar to those shown in the remainder of this paper are available through the web interface of the DROUGHT-HEAT Regional Climate Atlas. All plots available through this interactive interface are based
on the computation of the empirical dependency relationship using the S16 framework as described in the previous section.

The layout and individual components of the DROUGHT-HEAT Regional Climate Atlas are shown in Fig. 2. Plots are drawn by making the appropriate selections in the data panel (left-hand side of the screen shot). The first item to select is the diagnostic (i.e., "Dependency with global mean temperature" for the results of this study). After that, the data source drop-down is populated with a list of available data sets (i.e., CMIP5 model simulations for this study, for either the period 1861–2099
or 1861–2299). Equivalently, the drop-down labelled "Select Index or Variable" is filled with available indices. Credits of the selected diagnostic and data source are displayed on the right-hand panel.

The map in the data panel shows the set of SREX regions that can be chosen for the scaling analyses. Other region sets can be selected by using the drop-down on top of the map (e.g., also further regions used in S16, such as the contiguous US, central Brazil, the Arctic and Southern Asia). Once the user has selected a region (by either clicking on one of the polygons in the
15 map or by selecting a global domain), the requested plot is displayed in the main panel of the website. When the appropriate selections are made, a link appears allowing the user to navigate to a set of box plots showing the distribution of the selected index for fixed global mean temperature targets of $1.5°C$, $2°C$ and $3°C$ (see more details in Sect. 3.2).

The atlas has been designed to be self-explanatory. Each item in the drop-down lists is accompanied by a short help text that shows up when hovering over it with the mouse. In addition, a pop-up window has been added providing help for first-time
users.

#### 2.4.2 Technical implementation of the plotting framework

The DROUGHT-HEAT Regional Climate Atlas is based on a number of web modules served through the Gunicorn web application server (http://gunicorn.org/) and the NGINX reverse-proxy server (https://www.nginx.com/). The website is built within the Django web framework (https://www.djangoproject.com/). It is hosted on a web server at ETH Zurich.
The map shown in the data panel of the DROUGHT-HEAT Regional Climate Atlas (see Fig. 2) is based on Leaflet (http://leafletjs.com/). The background (world) layer is based on tilesets served via Mapbox (https://www.mapbox.com/). The region boundaries are read from text files in GeoJSON format.

There are two processing layers required to produce plots within the framework. First, a locally hosted ncl script serves static comma-separated values (CSV) files to the web server. The script writes the data points of each plot series into files
inside a unique folder which represents the diagnostic, region and index. It also generates two customizable files containing plot and series configuration parameters for each index. In the second (server-sided) layer, the csv files are read and processed by JavaScript code. Finally, the Highcharts charting library (http://www.highcharts.com/) parses the input files to generate the desired plot.





## 3   Results and discussion

In the following, we demonstrate the capabilities of the DROUGHT-HEAT Regional Climate Atlas by presenting some selected results. We also discuss some more in-depth analyses considering specific features of the assessed dependency relationships between regional climate and global temperature changes.

### 3.1   Dependency relationships

Figure 3 displays the relations of regional changes in temperature based climate and extremes indices in various SREX regions to global mean temperature ($\Delta T_{glob}$). The indices show an apparent linear scaling with $\Delta T_{glob}$ when solely considering the ensemble mean change (the significance of the relationship of individual ensemble members is tested below, see Table 3). As all indices in Fig. 3 are derived from temperatures, the scaling of changes in these indices shows similar linear features than the scaling of changes in regional mean temperatures ($\Delta T$, first row of Fig. 3). Moreover, the dependency relationship of these indices involves the least uncertainties when compared to the other indices shown in Fig. 4. For all of the indicated regions, the slope of the temperature-based indices is consistently above one (although only by a small margin for $\Delta TXn$ in the Amazon region, AMZ), indicating a larger change of the regional indices compared to $\Delta T_{glob}$. For instance at 2 °C global warming, the warming in hot extremes (annual maximum of the daily maximum temperature, $TXx$) in the Mediterranean (SREX region MED) amounts to 3.2 °C. The largest departures from the identity line are found for changes in the annual minimum of both daily maximum and minimum temperatures ($\Delta TXn$ and $\Delta TNn$) in NEU.

For the precipitation-based indices discussed here, the responses are often less pronounced and subject to larger inter-model uncertainties (Fig. 4). Nevertheless, the ensemble mean changes of the purely precipitation based indices ($\Delta P$, $\Delta Rx5day$, $\Delta CDD$ and $\Delta SPI12$) still show a significant linear scaling with $\Delta T_{glob}$ in some regions. For example, there is a clear tendency for a positive scaling of heavy precipitation ($\Delta Rx5day$) with $\Delta T_{glob}$ in Central Europe (CEU), North Europe (NEU), Central North America (CNA) and East Asia (EAS). Moreover, MED displays a remarkable increase in the maximum dry spell lengths ($\Delta CDD$) by the end of the century (i.e., the decade in which global temperature anomalies are projected to reach $\Delta T_{glob} = 4.75°C$ in the RCP8.5 scenario). This is consistent with the response of the drought indices ($\Delta SPI12$, $\Delta SMA$ and $\Delta P - E$) in this region towards drying, although the large uncertainties in $\Delta SMA$ near the end of the century must not be ignored. Apart from the positive scaling of $\Delta SPI12$ in NEU and EAS and the wetting signal indicated by $\Delta P - E$ in NEU, the responses are connected with large uncertainties and both an increase and a decrease of these indices is within the projected range even for large values of $\Delta T_{glob}$. Note that the differences in between the scaling of mean precipitation and heavy precipitation could possibly be explained by different sensitivities to aerosol loading (Pendergrass et al., 2015).

Overall, the dependency relationship is very similar for the four emission scenarios (Figures 3 and 4). Thus regional changes in the indicated indices can be usefully related to given cumulative $CO_2$ targets, independently of the emission pathway.

Table 3 displays the significant linear trends of the previously discussed indices of the RCP8.5 scenario for $\Delta T_{glob} \geq 1°C$. Models generally agree that changes in global mean temperatures translate into enhanced changes both in regional mean temperatures over land as well as in regional temperature extremes. The scaling with precipitation-derived indices shows a



much more diverse pattern. Heavy precipitation events (as reflected by $Rx5day$) are projected to intensify over several of the selected regions, most strikingly over NEU, EAS and EAF (East Africa). Dry spells are projected to become longer mainly over MED and AMZ, which is in line with both a decrease in precipitation and enhanced soil moisture depletion as shown by $\Delta SMA$ (although projections of CDD are generally dominated by larger uncertainties, which is in part due to high model sensitivities

5  related to the binary cut-off of $1mm$ used to distinguish dry days from days with precipitation). The Mediterranean region (MED) is the only region for which all relevant indices point towards a distinct drying. In contrast, precipitation is projected to increase with increasing global mean temperatures over NEU, EAS and EAF. While this signal is consistent with the trend in $SPI12$ in each of the three regions, soil moisture anomalies are projected to only increase in EAF. Apart from MED, the model agreement on trends in $P - E$ is mostly poor.

## 3.2   1.5 °C vs. 2 °C response

Figures 5 and 6 present the CMIP5-based distributions of the changes in the various indices for 1.5 °C, 2 °C and 3 °C global warming, and for the four emission scenarios (appendix A shows the same type of plots for all other SREX regions as well as for global land, not discussed). Significance of the differences ($1.5°C$ vs. $2°C$) was assessed using a two-sided paired Wilcoxon test ($p = 0.01$, after controlling the false discovery rate according to Benjamini and Hochberg, 1995). Significant

differences between $1.5°C$ and $2°C$ global warming are observable for virtually all of the temperature-based indices, when excluding the RCP2.6 scenario (where only 6 out of 18 models reach $\Delta T_{glob} = 2°C$). For the precipitation-based indices, the differences in the response between the two global temperature targets is mostly insignificant. MED is projected to experience the strongest drying, as indicated by the significant increase in $\Delta CDD$ (RCP4.5 and RCP8.5) and the corresponding decrease in water availability, as reflected by the decrease in $\Delta SMA$ (RCP8.5) and a decrease in $\Delta P - E$ (RCP4.5), confirming that

this region is a potential hot spot for future drought-related changes (Orlowsky and Seneviratne, 2013; Guiot and Cramer, 2016; Schleussner et al., 2016). On the other hand, NEU and EAS (Fig. 6) experience a significant increase in wet extremes. The other non-temperature indices show mostly no statistically significant distinction in the response between the two global temperature targets. The large spread in the precipitation based indices in AMZ indicates that precipitation projections in this region are subject to substantial uncertainties.

## 3.3   Intra-model variability

The dependency relationships and uncertainty ranges discussed so far are based on one ensemble member (r1i1p1) of the applied models (see Table 1). In order to investigate any impact of intra-model variability on this range, Fig. 7 displays the ensemble mean and uncertainty ranges based on all ensemble members available for each model vs. the one-member based ensemble mean and uncertainty range on the example of the precipitation-based indices discussed earlier. The regional signal

of the dependency relationship of $\Delta SMA$ based on all ensemble members of the RCP4.5 scenario shows some inconsistencies with the other scenarios, which was found to be due to biases in an individual model simulation. Apart from this, the consideration of all members (and thus intra-model variability) results in a marginally enhanced uncertainty range (the largest enhancements were found for $\Delta CDD$ and $\Delta SMA$), while the ensemble mean is nearly identical to the ensemble mean of the





one-member based indices. Thus the uncertainty ranges based on one member seem to be appropriate to also cover intra-model variability. However, a number of models provide only the r1i1p1 simulation, potentially resulting in an underestimation of the true inter-model variability. Moreover, including only one run per model avoids that models which provide more runs have a higher weight in the ensemble results.

## 3.4   Beyond 2100

While most CMIP5 model simulations end by the end of the twenty-first century, a few simulations are available up to the year 2299 (see Table 1). These allow us to analyse the scaling relationship beyond 2100 and to assess their longer-term behaviour.

The long-term dependency of changes in temperature-related indices on changes in global temperature is similar (i.e., mostly linear in the ensemble mean) to the one shown in Fig. 3 (not shown). For the other indices, the linear scaling assumption for

the 1861–2299 period apparently only holds for a subset of indices and differs among regions (Fig. 8). Regions in which the indices scale linearly with $\Delta T_{glob}$ in the RCP8.5 scenario are also often characterized by a near-linear response in the other scenarios, which is remarkable given the fact that $\Delta T_{glob}$ is projected to remain constant or to decrease over time in these scenarios. In the RCP8.5 scenario, the trend towards more extreme dry conditions in MED (and partly in AMZ) is projected to continue also beyond 2100, while NEU and EAS are characterized by a continuation of the intensification in wet extremes

on both short ($\Delta Rx5day$) and longer-term ($\Delta SPI12$) time scales. Irrespective of the changes in $\Delta P$, $\Delta Rx5day$ continues to increase in a near-linear fashion in all regions except MED.

## 4   Conclusions

We have developed the "DROUGHT-HEAT Regional Climate Atlas", a new interactive web interface available via the URL http://www.drought-heat.ethz.ch/atlas, which provides dependency relationships between changes in regional climate indices

and global mean temperature for 26 larger IPCC pre-defined regions. Beside acting as a platform to foster scientific discussion, the aim of this web interface is to increase the accessibility of peer-reviewed scientific results to the general public, which is of major concern for the communication of climate science findings (e.g., Harold et al. 2016). This is particularly relevant for the critical evaluation of the regional-scale implications of considered global temperature limits, such as the $1.5°C$ and $2°C$ temperature goals established in the 2015 Paris Agreement.

With the selected results presented here, we have demonstrated that a number of regionally averaged climate indices show a distinct linear relationship with global mean temperatures both in the ensemble mean and in individual CMIP5 model realizations, as also illustrated in S16 for a more limited set of indices and emissions scenarios. The linear relationship is particularly obvious for the analysed temperature-derived indices, and still present for a number of drought and water cycle indices. We note, however, that some analyses display departures from such linear relationships, in particular in the case of indices showing

a low signal to noise in projections (e.g. in several regions for mean precipitation, dry spell lengths, soil moisture anomalies, and precipitation minus evapotranspiration). Such departures are generally more pronounced in the RCP2.6 scenario, because of the weak overall forcing in that emission scenario, and possibly also because of differences in aerosol forcing in RCP2.6

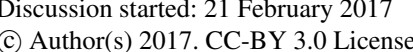


compared to the other emission scenarios (Pendergrass et al. 2015). These cases of non linearities illustrate the advantage of the applied S16 approach compared to traditional pattern scaling approaches, as the derived dependencies are purely empirical and not assessed from a priori determined mathematical relationships.

Projected changes in the indices are overall larger in a $2°C$ world (i.e., $\Delta T_{glob} = 2°C$ relative to pre-industrial levels) compared to a $1.5°C$ world (i.e., $\Delta T_{glob} = 1.5°C$ relative to pre-industrial levels). The differences between the two global temperature limits are particularly large and generally significant for regional mean and extreme temperatures. Results tend to be less robust for water-cycle indices, in particular for those related to water availability (soil moisture anomalies or precipitation minus evapotranspiration). We encourage the reader to use the DROUGHT-HEAT Regional Climate Atlas to evaluate these regional dependency relationships using other indices or other regions than those presented in this study.

The DROUGHT-HEAT Regional Climate Atlas has been designed to be easily expanded both in terms of functionality (e.g., adding support for additional plot types) and in terms of the number and type of supported data sets and diagnostics. By these means we facilitate an easy extension of the platform to include graphical material from upcoming publications within the scope of the DROUGHT-HEAT project and beyond.

## 5 Code availability

All code used to prepare the results discussed within this study is available upon request from the first author.

## 6 Data availability

All data produced within this study is available via the website http://drought-heat.ethz.ch/atlas/ through the export functions of the plots, or upon request from the first author.

## Appendix A: 1.5 °C vs. 2 °C response

## 7 Competing interests

The authors declare that they have no conflict of interest.

*Acknowledgements.* R.W. and S.I.S acknowledge the European Research Council (ERC) 'DROUGHT-HEAT' project funded by the European Community's Seventh Framework Programme (grant agreement FP7-IDEAS-ERC-617518). This study contributes to the World Climate Research Programme (WCRP) Grand Challenge on Extremes. We acknowledge the World Climate Research Programme's Working Group on Coupled Modelling, which is responsible for CMIP, and we thank the climate modelling groups (listed in Table 1 of this paper) for producing and making available their model output. For CMIP the U.S. Department of Energy's Program for Climate Model Diagnosis and Intercomparison provides coordinating support and led development of software infrastructure in partnership with the Global Organization for Earth System Science Portals.



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





**Table 1.** List of models used in this study (in alphabetical order). Crosses (circles) indicate availability of simulations of the ensemble member r1i1p1 for the 1861–2099 (1861–2299) period. Note that the number of simulations of other ensemble members is considerably smaller.

| Model | Modelling Centre | Historical | RCP2.6 | RCP4.5 | RCP6.0 | RCP8.5 |
|---|---|---|---|---|---|---|
| ACCESS1-0 | Commonwealth Scientific and Industrial Research Organization (CSIRO) and Bureau of Meteorology (BOM), Australia | x | | x | | x |
| bcc-csm1-1 | Beijing Climate Center, China Meteorological Admin- | x | o | o | x | o |
| bcc-csm1-1-m | istration | x | x | x | x | x |
| CanESM2 | Canadian Centre for Climate Modelling and Analysis | x | x | x | | x |
| CCSM4 | National Center for Atmospheric Research, USA | x | | o | o | o |
| CMCC-CM | Centro Euro-Mediterraneo per i Cambiamenti Cli- | x | | x | | x |
| CMCC-CMS [a] | matici, Italy | x | | x | | x |
| CNRM-CM5 | Centre National de Recherches Météorologiques / Centre Européen de Recherche et Formation Avancées en Calcul Scientifique, France | x | x | x | | x |
| CSIRO-Mk3-6-0 | Commonwealth Scientific and Industrial Research Organization / Queensland Climate Change Centre of Excellence, Australia | x | x | o | x | o |
| FGOALS-s2 | LASG, Institute of Atmospheric Physics, Chinese Academy of Sciences | x | x | x | x | x |
| GFDL-CM3 | | x | x | | x | x |
| GFDL-ESM2G | NOAA Geophysical Fluid Dynamics Laboratory, USA | x | x | x | x | x |
| GFDL-ESM2M | | x | x | x | x | x |
| HadGEM2-CC | Met Office Hadley Centre, United Kingdom | x | | x | | x |
| HadGEM2-ES | | x | o | | x | o |
| inmcm4 | Institute for Numerical Mathematics, Russia | x | | x | | x |
| IPSL-CM5A-LR | | x | o | o | x | o |
| IPSL-CM5A-MR | Institut Pierre-Simon Laplace, France | x | x | | x | x |
| IPSL-CM5B-LR | | x | | x | | x |
| MIROC-ESM | Japan Agency for Marine-Earth Science and Technology, Atmosphere and Ocean Research Institute (The University of Tokyo), and National Institute for Environmental Studies | x | x | x | x | x |
| MIROC-ESM-CHEM | | x | x | x | x | x |
| MIROC5 | | x | x | x | x | x |
| MPI-ESM-LR | Max Planck Institute for Meteorology, Germany | x | o | o | | o |
| MPI-ESM-MR | | x | x | x | | x |
| MRI-CGCM3 | Meteorological Research Institute, Japan | x | x | x | x | x |
| NorESM1-M | Norwegian Climate Centre | x | x | x | x | x |

[a] not used for calculation of $P - E$



**Table 2.** List of indices (in alphabetical order) as presented in the DROUGHT-HEAT DROUGHT-HEAT Regional Climate Atlas. Crosses denote indices specifically discussed in this paper as well as indices expressed as percent changes relative to the pre-industrial reference period 1861–1880.

| Index | Description | Unit | Expressed as % change | Discussed in this paper | Reference for computation |
|---|---|---|---|---|---|
| CDD | Maximum length of dry spell | days | | x | Sillmann et al., 2013a, b |
| CSDI | Cold speel duration index | days | | | Sillmann et al., 2013a, b |
| CWD | Maximum length of wet spell | days | | | Sillmann et al., 2013a, b |
| DTR | Daily temperature range | °C | | | Sillmann et al., 2013a, b |
| FD | Number of frost days | days | | | Sillmann et al., 2013a, b |
| GSL | Growing season length | days | | | Sillmann et al., 2013a, b |
| ID | Number of icing days | days | | | Sillmann et al., 2013a, b |
| P-E | Precipitation - evapotranspiration | mm/day | | x | Greve and Seneviratne, 2015 |
| P | Mean precipitation | mm | x | x | Taylor et al., 2012 |
| PRCPTOT | Annual total precipitation in wet days | mm | x | | Sillmann et al., 2013a, b |
| R10mm | Annual count of days when PRCP ≥ 10mm | days | | | Sillmann et al., 2013a, b |
| R1mm | Annual count of days when PRCP ≥ 1mm | days | | | Sillmann et al., 2013a, b |
| R20mm | Annual count of days when PRCP ≥ 20mm | days | | | Sillmann et al., 2013a, b |
| R95pTOT | Annual total PRCP when RR > 95p. | mm | | | Sillmann et al., 2013a, b |
| R99pTOT | Annual total PRCP when RR > 99p. | mm | | | Sillmann et al., 2013a, b |
| Rx1day | Monthly maximum 1-day precipitation | mm | x | | Sillmann et al., 2013a, b |
| Rx5day | Monthly maximum 5-day precipitation | mm | x | x | Sillmann et al., 2013a, b |
| SDII | Simple precipitation intensity index | mm/day | | | Sillmann et al., 2013a, b |
| SMA | Soil moisture anomalies | 1 | | x | Orlowsky and Seneviratne, 2013 |
| SPI12 | Standardized precipitation index (12-month accumulation period) | 1 | | x | Vicente-Serrano et al., 2010 |
| SU | Number of summer days | days | | | Sillmann et al., 2013a, b |
| T | Mean temperature | °C | | x | Taylor et al., 2012 |
| TN10p | Percentage of days when TN < 10th percentile | % days | | | Sillmann et al., 2013a, b |
| TN90p | Percentage of days when TN > 90th percentile | % days | | | Sillmann et al., 2013a, b |
| TNn | Monthly minimum of daily min. temperature | °C | | x | Sillmann et al., 2013a, b |
| TNx | Monthly maximum of daily min. temperature | °C | | x | Sillmann et al., 2013a, b |
| TR | Number of tropical nights | days | | | Sillmann et al., 2013a, b |
| TX10p | Percentage of days when TX < 10th percentile | % days | | | Sillmann et al., 2013a, b |
| TX90p | Percentage of days when TX > 90th percentile | % days | | | Sillmann et al., 2013a, b |
| TXn | Monthly minimum of daily max. temperature | °C | | x | Sillmann et al., 2013a, b |
| TXx | Monthly maximum of daily max. temperature | °C | | x | Sillmann et al., 2013a, b |
| WSDI | Warm spell duration index | days | | | Sillmann et al., 2013a, b |





**Figure 2.** Screen-shot of the DROUGHT-HEAT Regional Climate Atlas. For demonstration, this screen-shot displays the scaling plot of $\Delta T_{glob}$ against $\Delta TXx$ based on model simulations from 1861–2099 for the SREX region West North America (WNA).





**Figure 3.** Scaling plots for the indices $\Delta T$, $\Delta TXx$, $\Delta TXn$, $\Delta TNx$, and $\Delta TNn$, based on CMIP5 simulations of ensemble member r1i1p1 and averaged over the SREX regions MED, CEU, NEU, CNA, EAS and AMZ.





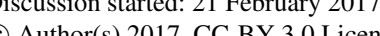

**Figure 4.** Like Fig. 3, but for the indices $\Delta P$, $\Delta Rx5day$, $\Delta CDD$, $\Delta SPI12$, $\Delta SMA$, and $\Delta P - E$. Values for $\Delta CDD > 100 days$ were cut off for readability.



**Table 3.** Scaling slopes of the RCP8.5 scenario for $\Delta T_{glob} \geq 1^\circ C$ and percent of models with a statistically significant linear scaling (in brackets, $p = 0.01$) for various SREX regions, based on CMIP5 simulations of ensemble member r1i1p1. Bold values indicate significance for at least $50\%$ of the contributing models for which the sign of the trend is identical to the sign of the ensemble mean trend.

| Index | Regions | | | | | | | |
|---|---|---|---|---|---|---|---|---|
| | MED | CEU | NEU | CNA | AMZ | EAS | SAU | EAF |
| $\Delta T/\Delta T_{glob}$ [°C/°C] | **1.24** (100) | **1.26** (100) | **1.36** (100) | **1.39** (100) | **1.30** (100) | **1.36** (100) | **0.96** (100) | **1.15** (100) |
| $\Delta TXx/\Delta T_{glob}$ [°C/°C] | **1.65** (100) | **1.77** (100) | **1.30** (96) | **1.66** (100) | **1.59** (100) | **1.49** (100) | **1.10** (100) | **1.19** (100) |
| $\Delta TXn/\Delta T_{glob}$ [°C/°C] | **1.11** (100) | **2.00** (100) | **2.53** (100) | **1.86** (100) | **1.03** (100) | **1.45** (100) | **0.85** (100) | **0.92** (100) |
| $\Delta TNx/\Delta T_{glob}$ [°C/°C] | **1.55** (100) | **1.52** (100) | **1.21** (100) | **1.48** (100) | **1.52** (100) | **1.34** (100) | **1.08** (100) | **1.24** (100) |
| $\Delta TNn/\Delta T_{glob}$ [°C/°C] | **1.11** (100) | **2.35** (100) | **2.77** (100) | **2.05** (100) | **1.24** (100) | **1.61** (100) | **0.81** (100) | **1.28** (100) |
| $\Delta P/\Delta T_{glob}$ [%/°C] | **-5.87** (65) | 0.62 (23) | **4.53** (92) | 0.93 (12) | -1.67 (23) | **4.13** (92) | -2.10 (8) | **5.38** (65) |
| $\Delta Rx5day/\Delta T_{glob}$ [%/°C] | -0.83 (8) | **3.59** (85) | **5.10** (100) | **3.42** (77) | **3.20** (73) | **6.52** (100) | 2.10 (12) | **7.73** (85) |
| $\Delta CDD/\Delta T_{glob}$ [days/°C] | **10.73** (96) | 1.31 (50) | 0.15 (12) | 0.68 (8) | **3.78** (62) | -0.91 (31) | **2.75** (58) | -0.93 (8) |
| $\Delta SPI12/\Delta T_{glob}$ [1/°C] | **-0.32** (72) | 0.05 (28) | **0.35** (92) | 0.06 (20) | -0.15 (36) | **0.23** (92) | -0.10 (24) | **0.22** (64) |
| $\Delta SMA/\Delta T_{glob}$ [1/°C] | **-0.62** (88) | -0.07 (28) | -0.12 (28) | -0.13 (36) | **-0.36** (52) | -0.10 (40) | -0.01 (28) | **0.34** (68) |
| $\Delta P - E/\Delta T_{glob}$ [mm/°C] | **-0.05** (80) | -0.01 (8) | **0.05** (56) | -0.00 (0) | -0.04 (16) | 0.03 (32) | -0.01 (8) | 0.05 (40) |



**Figure 5.** Response of indices $\Delta T$, $\Delta TXx$, $\Delta TXn$, $\Delta TNx$, $\Delta TNn$, $\Delta P$, $\Delta Rx5day$, $\Delta CDD$, $\Delta SPI12$, $\Delta SMA$ and $\Delta P - E$ to a global temperature increase of $1.5^\circ C$, $2^\circ C$ and $3^\circ C$, based on CMIP5 simulations of ensemble member r1i1p1 and averaged over the European SREX regions MED, CEU and NEU. The upper and lower hinges of the box plots represent the first and third quartile. The whiskers extend to the highest (lowest) value that is within 1.5 times the interquartile range of the upper (lower) hinge. Values outside this range are displayed as dots.





**Figure 6.** Like Fig. 5, but for SREX regions CNA, AMZ and EAS.





**Figure 7.** Dependency relationships for the indices $\Delta P$, $\Delta Rx5day$, $\Delta CDD$, $\Delta SPI12$, $\Delta SMA$, and $\Delta P - E$, averaged over the SREX regions MED, CEU and NEU. Indices based on all CMIP5 ensemble members available per model (solid lines, dark shading) are compared with indices based on ensemble member r1i1p1 of each model (dashed lines, light shading). Values for $\Delta CDD > 100\,days$ and $\Delta SMA < -5$ were cut off for readability.



**Figure 8.** Like Fig. 4, but for the time period 1861–2299. Values for $\Delta CDD > 200\,days$ were cut off for readability.





**Figure A1.** Response of indices $\Delta T$, $\Delta TXx$, $\Delta TXn$, $\Delta TNx$, $\Delta TNn$, $\Delta P$, $\Delta Rx5day$, $\Delta CDD$, $\Delta SPI12$, $\Delta SMA$ and $\Delta P - E$ to a global temperature increase of $1.5°C$, $2°C$ and $3°C$, based on CMIP5 simulations of ensemble member r1i1p1 and averaged over the SREX regions ALA, CAM and CAS. The upper and lower hinges of the box plots represent the first and third quartile. The whiskers extend to the highest (lowest) value that is within 1.5 times the interquartile range of the upper (lower) hinge. Values outside this range are displayed as dots.





**Figure A2.** Like Fig. A1, but for SREX regions CGI, EAF and ENA.





**Figure A3.** Like Fig. A1, but for SREX regions NAS, NAU and NEB.





**Figure A4.** Like Fig. A1, but for SREX regions SAF, SAH and SAS.





**Figure A5.** Like Fig. A1, but for SREX regions SAU, SEA and SSA.





**Figure A6.** Like Fig. A1, but for SREX regions TIB, WAF and WAS.





**Figure A7.** Like Fig. A1, but for SREX regions WNA, WSA and for global land ("Global").