# Peer review of "Changes in regional climate extremes as a function of global mean temperature: an interactive plotting framework"

_Geoscientific Model Development, 2017_

## Referee Comment (RC1) · B. Sanderson (Referee) · 3 Jun 2017

The submitted manuscript outlines a tool, based on the prior work of Seneviratne et al (2016), which provides a convenient visualization for regional climate extremes changes as a function of global mean temperatures. The paper serves as documentation for an accompanying online tool, which allows end users to quickly produce regional relevant climate data.

The paper is well written, and the web-based tool appears to nicely perform the functions advertised in the paper - which will doubtless be very useful for end users as an easily accessible source of multi-model climate data. As such, the paper provides a

nice resource for the wider community, and should certainly be published as a result.

As a scientific paper - the results are accurate, and though they are not groundbreaking - this is clearly a paper whose major function is to provide documentation for a useful tool, and so the results serve primarily as illustrations of the tool's capabilities.

Minor points: - The breakdown of the importance of multi-model variability and internal variability is not particularly meaningful. The authors compare an ensemble of r1i1p1 members, with an ensemble using all initial condition members from all models. Unsurprisingly, the results are very similar, as one would expect both ensembles to be subject to both initial condition and structural uncertainty. Perhaps a more meaningful comparison would be to look at the spread in initial condition members from a single model version, where a significant sample is available - this would give a more meaningful interpretation of what fraction of the multi-model spread is due to initial condition variability alone.

Section 3.1: would the authors like to comment a little on potential mechanisms for why different regions exhibit different gradients of response. The NEU response seems likely associated with polar amplification. Is the Amazon response typical of the tropics in general - or is there a particular feedback associated with this region?

- Could the authors comment a little on how sensitive the 1.5 / 2 degree results are to the scenario chosen? Establishing whether there are significant differences in the distributions of response using different scenarios in any regions arising from the method would be a useful result for the pattern scaling community which could be trivially assessed from the analysis already done here.

- the shaded regions in Figures 7/8 are a little confusing, and could do with a little more explanation. Presumably - discontinuities arise because some models never reach some levels of warming in some scenarios, but the fact that the number of models change along the x-axis makes the grey bars difficult to interpret. Are some of the apparent nonlinearities mainly due to the fact that the ensemble sample is changing

along the x-axis?

---

## Referee Comment (RC2) · Anonymous Referee #2 · 4 Jun 2017

The paper describes a very useful tool to assess regional changes in extreme indices in term s of global mean temperature change. The paper is clearly written und should be punblished with minor modifications addressing the issues and questions below.

Main point:

The relationship between ensemble mean changes in extreme indicators in term so global mean temperature change may partly been due to changes in the ensemble itself as not all of the simulation reach the same level of warming. It is possible to illustrate this by showing the number of data points in the individual samples as a histogram to the x-axis of the plots?

[Figure]

Minor points:

It is quite relevant that the data is not only visualized but also available for download. That seems to be the case as mentioned only at the end of the manuscript. It should be highlighted earlier in the text. The data associated with the ensemble plots should also contain the information about each individual model run.

P2, L21: "the probability of exceeding a given temperature threshold" instead of "exceedance of a given temperature threshold"

P5, L1: "Derivation of the relationship between changes in regional climate indices and global mean temperature" instead of "Derivation of regional global temperature dependency relationships"?

P5, L5: Throughout the manuscript "predict" or "prediction" should be replaced by "project" or "projection".

P5, L13: Throughout the manuscript "global temperature" should be replaced by "global mean temperature" and "dependency relationship" could be replaced by "(functional) relationship". In this case I would suggest to write "To test the significance of the relationship between global mean temperature change and the regionally averaged..."

P5, L16: I think it is more precise to say "The number of models for which the slope of the regression line is significantly different from zero..."

P5, L19-21: I do not think that this is true. A scaling coefficient that is significantly different from zero does not imply that the relationship can really be explained by a linear model. That is an assumption made when fitting the data. The approach only tests whether the scaling coefficient is significantly different from zero assuming that the data follow the underlying linear model.

P5, L22: Are the decadal averages not actually calculated to allow for the averaging?

P7, L5: I would suggest do name it "Functional form of the relationship" (see comment

above)

P7, L9: I think it is "to" instead of "than".

P7, L10-11: "Moreover, the relationship of these indices involves the least uncertainties..." How is that measured?

P7, L15-16: Similarly, "The largest departures from the identity line..." How is that measured?

P7, L18: "The ensemble mean changes... still show a significant linear scaling...". How is that tested? In the method section that is only described for individual runs while the entire ensemble has a more complex structure where e.g. results from different scenarios but one climate models cannot be considered as independent. Even if it is tested in this simple way it should be described in the method section and the limitations should be mentioned.

P7, L27: Delete "in". P7, L29-30: How is that derived? It should be explained as it seems to be part of the plot shown in Figure 2.

P8, L13-14: This description should be moved to section 2.3. Is the test only applied to individual scenarios, i.e. a sample where containing one element per climate model, or also across the four scenarios, i.e. a larger sample with up to four data point per model? In the latter case the elements of the sample cannot be considered independent. How is that addressed?

P8, L31: Change "an" to "one".

P3, Figure 1: The Caption should include the definition of TXx.

P16, Figure 2: Could the different uncertainty ranges be shown in transparent colors that one of them does not hide the other one? Would be nice to add a histogram to illustrate the number of data points behind the lines (see first comment). It has to be explained how the CO2 budgets are derived. That should be described in the main

Interactive
comment

text. In the caption the term "CO2 emissions" should be adjusted. It is not precise enough.

P17, Figure 3: The caption should include a reference to Table 2 where the indices are described. Why does the shaded area extent beyond the line of the ensemble mean?

P20, Figure 5: The caption should also say that the cetral line of the box plot is represents the median.

P22, Figure 7: Could the different uncertainty ranges be shown in transparent colors? Here, it would also be good to add the information about the number of data points (or models) included in the samples to get a better idea whether non-linearities in mean response could also be due to changes in the underlying samples.

P23, Figure 8: Why does the shaded area extent beyond the line of the ensemble mean?

---

## Author Comment (AC1) · 26 Jun 2017

**Comments from referee RC1 and author's response**

**Minor points**

*- The breakdown of the importance of multi-model variability and internal variability is not particularly meaningful. The authors compare an ensemble of r1i1p1 members, with an ensemble using all initial condition members from all models. Un- surprisingly, the results are very similar, as one would expect both ensembles to be subject to both initial condition and structural uncertainty. Perhaps a more meaningful comparison*

*would be to look at the spread in initial condition members from a single model version, where a significant sample is available - this would give a more mean- ingful interpretation of what fraction of the multi-model spread is due to initial condition variability alone.*

We choose to show the ensemble spread as a combined effect of structural uncertainty and initial conditions from all models, as we would like to show the magnitude by which the overall spread (based on all models) is influenced by the number of model runs used. We could of course also pick a single model with a sufficient number of real- izations, but then we would not be able to say whether the magnitude of this spread is a good estimate for internal variabiliy of models with an insufficient number of initial condition members nor could we say whether this spread dominates the magnitude of the inter-model ensemble spread or not.

*- Section 3.1: would the authors like to comment a little on potential mechanisms for why different regions exhibit different gradients of response. The NEU response seems likely associated with polar amplification. Is the Amazon response typical of the tropics in general - or is there a particular feedback associated with this region?*

Thanks for these suggestions. Although we do not intend to substantially extend the interpretation of our results, we added a few extra statements about the responses found in NEU and AMZ (see tracked changes document).

*- Could the authors comment a little on how sensitive the 1.5 / 2 degree results are to the scenario chosen? Establishing whether there are significant differences in the dis- tributions of response using different scenarios in any regions arising from the method would be a useful result for the pattern scaling community which could be trivially as- sessed from the analysis already done here.*

We agree that these results may be of interest for the pattern scaling community. We added two sentences (for the temperature and the precipitation based indices) stating by what degree the significance of the differences of 1.5 vs. 2 degree is dependent on

the scenario (see tracked changes document).

*- the shaded regions in Figures 7/8 are a little confusing, and could do with a little more explanation. Presumably - discontinuities arise because some models never reach some levels of warming in some scenarios, but the fact that the number of models change along the x-axis makes the grey bars difficult to interpret. Are some of the apparent nonlinearities mainly due to the fact that the ensemble sample is changing along the x-axis?*

The reviewer is right, the apparent discontinuities in the shaded areas arise from individual models that do not reach a specific level of warming. We still prefer to show $\Delta T_{glob}$ of up to $6°C$ to also include models that predict very strong warming and to see the entire range. The spread always has to be interpreted as a spread in both $\Delta T_{glob}$ and $\Delta I$. This is explained on page 5 lines 26f. We were asked by reviewer 2 to add a Figure showing the number of models simulating specific levels of global mean temperature, which we have added to the appendix.

**Author's changes**

Please find attached a marked-up version of the manuscript with all changes highlighted.

Please also note the supplement to this comment:
https://www.geosci-model-dev-discuss.net/gmd-2017-33/gmd-2017-33-AC1-supplement.pdf

---

## Author Comment (AC2) · 26 Jun 2017

**Comments from referee RC2 and author's response**

**Main points**

*- The relationship between ensemble mean changes in extreme indicators in terms of global mean temperature change may partly been due to changes in the ensemble itself as not all of the simulation reach the same level of warming. It is possible to illustrate this by showing the number of data points in the individual samples as a histogram to the x-axis of the plots?*

[Figure]

This is certainly possible, but it would also require to extend Figures 3, 4, 7 and 8 by an additional row and make them thus less readable. We now provide a separate plot in the appendix (also attached below) showing the number of models in the ensemble, which can then be compared to any of the scaling plots shown in the paper or in the atlas, as the number of models used for each of the plots is independent from both the region and the index (apart from P-E, see Table 1) and only depends on the chosen length of the scenario period (1861-2099 vs. 1861-2299) and whether we use all runs or only r1i1p1.

**Minor points**

*- It is quite relevant that the data is not only visualized but also available for download. That seems to be the case as mentioned only at the end of the manuscript. It should be highlighted earlier in the text. The data associated with the ensemble plots should also contain the information about each individual model run.*

It is indeed possible to download the data associated with each plot. We added a short note in section 2.4.1. However, the down-loadable files only contain the information required for drawing the actual plot, i.e., they only contain the ensemble mean, min and max in case of scaling plots. We think that this is sufficient for most of the users of the atlas. An extension of the atlas to also provide the raw data associated with each plot, i.e. for each single model, lies beyond the current capabilities of the plotting framework that we have used.

*- P2, L21: "the probability of exceeding a given temperature threshold" instead of "exceedance of a given temperature threshold"*

Changed.

*- P5, L1: "Derivation of the relationship between changes in regional climate indices and global mean temperature" instead of "Derivation of regional global temperature dependency relationships"?*

We agree that this is more clear. Changed.

*- P5, L5: Throughout the manuscript "predict" or "prediction" should be replaced by "project" or "projection".*

Changed (found only one occurrence).

*- P5, L13: Throughout the manuscript "global temperature" should be replaced by "global mean temperature" and "dependency relationship" could be replaced by "(functional) relationship". In this case I would suggest to write "To test the significance of the relationship between global mean temperature change and the regionally averaged..."*

Although this wording is a bit heavy, we agree that it may be clearer to use the term "global mean temperature" and now use it throughout the revised manuscript. However, we decided to keep "dependency relationship", because the term "functional relationship" is too general and we want to stress that there is a dependency between global mean temperature and the regional mean indices.

*- P5, L16: I think it is more precise to say "The number of models for which the slope of the regression line is significantly different from zero..."*

Changed.

*- P5, L19-21: I do not think that this is true. A scaling coefficient that is significantly different from zero does not imply that the relationship can really be explained by a linear model. That is an assumption made when fitting the data. The approach only tests whether the scaling coefficient is significantly different from zero assuming that the data follow the underlying linear model.*

We have thoroughly discussed this point at time of writing, and included a statement in lines 20-22 to clarify it: "though it does not guarantee superiority of the linear model over other, higher-order polynomials". This makes clear that the linear model is not essentially the "best" model to explain the relationship, which is also not the purpose of the statistical test that we apply.

*- P5, L22: Are the decadal averages not actually calculated to allow for the averaging?*

We apply decadal averaging as we are interested in the longer-term signal and not the shorter-term fluctuations in the response of the indices. If the reviewer refers to spatial averaging, we note that this would also be possible with un-filtered data, so this is not the primary reason why we apply the running mean.

*- P7, L5: I would suggest do name it "Functional form of the relationship" (see comment above)*

We now write "Functional form of the dependency relationships".

*- P7, L9: I think it is "to" instead of "than".*

Changed.

*- P7, L10-11: "Moreover, the relationship of these indices involves the least uncertainties..." How is that measured?*

Measured by the ensemble spread. Changed.

*- P7, L15-16: Similarly, "The largest departures from the identity line..." How is that measured?*

Simply the departures of the ensemble mean from the identity line (also shown in the plot) - this should be trivial.

*- P7, L18: "The ensemble mean changes... still show a significant linear scaling...". How is that tested? In the method section that is only described for individual runs while the entire ensemble has a more complex structure where e.g. results from different scenarios but one climate models cannot be considered as independent. Even if it is tested in this simple way it should be described in the method section and the limitations should be mentioned.*

It is not tested here, we just wanted to describe that the linear relationship is visually

apparent. Replaced "significant" by "distinct".

*- P7, L27: Delete "in".*

OK

*P7, L29-30: How is that derived? It should be explained as it seems to be part of the plot shown in Figure 2.*

The $CO_2$ targets were derived in S16. Added reference.

*- P8, L13-14: This description should be moved to section 2.3. Is the test only applied to individual scenarios, i.e. a sample where containing one element per climate model, or also across the four scenarios, i.e. a larger sample with up to four data point per model? In the latter case the elements of the sample cannot be considered independent. How is that addressed?*

This test is only applied to individual scenarios. Moved to section 2.3.

*- P8, L31: Change "an" to "one".*

Changed.

*- P3, Figure 1: The Caption should include the definition of TXx.*

Added.

*- P16, Figure 2: Could the different uncertainty ranges be shown in transparent colors that one of them does not hide the other one? Would be nice to add a histogram to illustrate the number of data points behind the lines (see first comment). It has to be explained how the $CO_2$ budgets are derived. That should be described in the main text. In the caption the term "$CO_2$ emissions" should be adjusted. It is not precise enough.*

The uncertainty ranges are actually drawn in transparent colors (also in the online atlas). The light gray uncertainty band is either identical to the dark one or slightly

larger. I experimented with different colors but then there is too little focus on the colored lines depicting the scenario ensemble means.

We made the legend entry more precise by changing it to "Cumulative total CO2 emissions from 1870 (GtC) based on S16 and IPCC AR5 SPM figure 10". This change will be implemented in the public version of the atlas once the manuscript gets accepted.

*- P17, Figure 3: The caption should include a reference to Table 2 where the indices are described. Why does the shaded area extent beyond the line of the ensemble mean?*

Added. The shaded area extends beyond the line of the ensemble mean, as the data points on this line correspond to the ensemble mean of both $\Delta I_{reg}$ and $\Delta T_{glob}$, while $\Delta T_{glob}$ of the uncertainty range (gray shading) corresponds to the global mean temperature based on individual model simulations. This also explains why the ensemble mean can in some situations also appear outside the ensemble spread (e.g. $\Delta P - E$ in EAS, Figure 8).

*- P20, Figure 5: The caption should also say that the central line of the box plot is represents the median.*

Added.

*- P22, Figure 7: Could the different uncertainty ranges be shown in transparent colors? Here, it would also be good to add the information about the number of data points (or models) included in the samples to get a better idea whether non-linearities in mean response could also be due to changes in the underlying samples.*

The uncertainty ranges are already shown in transparent colors. The uncertainty range based on the r1i1p1 ensemble is (of course) the same or smaller than the range based on all members. We have added a separate Figure to the appendix showing the number of data points for both all ensemble members available and r1i1p1 only (see main point).

*- P23, Figure 8: Why does the shaded area extent beyond the line of the ensemble mean?*

See above.

**Author's changes**

Please find attached a marked-up version of the manuscript with all changes highlighted.

Please also note the supplement to this comment:
https://www.geosci-model-dev-discuss.net/gmd-2017-33/gmd-2017-33-AC2-supplement.pdf